# Does Urban and Peri-Urban Agriculture Contribute to Household Food Security? An Assessment of the Food Security Status of Households in Tongaat, eThekwini Municipality

**Nolwazi Zanele Khumalo [1] and Melusi Sibanda [2,\*]**

[1] Department of Agriculture, University of Zululand, KwaDlangezwa 3886, South Africa; zanellenkhumalo@gmail.com

[2] Faculty of Science and Agriculture, University of Zululand, KwaDlangezwa 3886, South Africa

\* Correspondence: SibandaM@unizulu.ac.za; Tel.: +27-(0)-35-902-6068

**Abstract:** Regardless of the steady increase in the economic growth of South Africa, poverty levels and food insecurity have not declined as one would have anticipated. Urban and peri-urban agriculture (UPA) presents an opportunity as a livelihood strategy to alleviate poverty and ensure household food security within the urban and peri-urban spheres. However, less research has been done in this area to discover the contribution of UPA on households' food security. This paper assesses the food security status of households that practised any form of UPA activities (later on referred to the rest of this paper as UPA practising households) and those that did not (later on referred to the rest of this paper as non-UPA practising households) within the Tongaat peri-urban area of eThekwini Municipality. The specific objectives of the paper are to estimate the household food security status of the UPA practising households vis-à-vis the non-UPA practising households and to elicit the reasons associated with the practice of UPA activities. Two hundred and eight (208) households (that is 109 and 99 UPA and non-UPA practising households respectively) were selected using a stratified random sampling procedure. The Household Dietary Diversity Score (HDDS) and Household Food Insecurity Access Score (HFIAS) measures were employed to estimate a household's food security status. A probit regression analysis was used to elicit the reasons associated with the practice of UPA activities by households. A non-parametric independent samples test (Mann-Whitney U) was used to compare whether there were significant differences between the two groups. A Pearson Chi-Square test reveals that the employment status, access to arable land, land tenure (ownership rights or arable land) and household monthly income variables were statistically significantly associated with the food security status (in terms of HDDS) of households. The results from the HDDS tool, show that a greater proportion (54%) of the UPA practising households consumed >6 food groups (deemed to be food secure in terms of dietary access) as compared to their counterparts, the non-UPA practising households (40%) in the same food group. However, the Mann-Whitney U test (U = 5292, $p$ = 0.808) show that there was no significant difference from this data in terms of the dietary diversity (HDDS) of the two groups. The HFIAS measure reveal that a greater proportion (about 72%) of the UPA practising households indicated that they never or rarely worried about food shortages (deemed to be food secure in terms of food access) as compared to their counterparts—the non-UPA practising households (about 61%) that never or rarely worried about food shortages. The Mann-Whitney U test (U = 4118.5, $p$ = 0.001) show that there was a significant difference in terms of food access (HFIAS) of the two groups. Overall, the results show that although UPA practising households seemed to be better off in terms of food access as compared to the non-UPA households, the results are inconclusive or show no evidence that a significant difference existed concerning dietary diversity of the two groups. The probit regression analysis shows that the variable arable land size ($p$ = 0.000) has a positive correlation with the practice of any UPA activity. In order for UPA to enhance the

household food security status (particularly in terms of dietary diversity) within the peri-urban spheres, an integrated approach (with agricultural land support from government and city planners) together with the diversification of high-value UPA activities by households is paramount.

**Keywords:** Household dietary diversity score; household food insecurity access scale; household food security; peri-urban agriculture; Tongaat

## 1. Introduction

The Food and Agriculture Organisation (FAO) of the United Nations (UN) and the World Bank set the eradication of hunger and poverty as their highest priority to be dealt with internationally [1]. The target date to achieve this goal and others was the year 2015 [2]. Various goals have been set out for countries profoundly affected by hunger, food insecurity and food shortages, matters relating to economic growth, equality in general and poverty. The significant movement being the Millennium Development Goals (MDGs), developed in 2000 at the United Nations Millennium Summit. South Africa, in line with the international community, has taken a stand in trying to reduce poverty as well as fulfil the other seven goals set out by the UN, by being part of the UN and making sure that the MDGs were achieved [2].

Nonetheless, the MDGs expired in 2015 and food insecurity including widespread poverty remains a challenge in South Africa. Numerous underprivileged South Africans are confronted with the increasing unemployment rate and therefore struggle to combat poverty eradication and food insecurity [3,4]. Although UPA is recognised as an essential livelihood strategy to curbing the presence of food insecurity within the urban and peri-urban areas, more research needs to be done to investigate the dynamics at play in the lives of those practising UPA [5]. Urban and peri-urban agriculture accounts for a substantial segment in the food supply of many capitals in sub-Saharan Africa [6]. By practising UPA, households can produce fresh milk, poultry products and vegetables. Urban and peri-urban agriculture is considered key to employment, poverty alleviation, livelihoods and a greater assortment of foods in the city marketplaces [7,8].

According to Briassoulis [9], urbanisation has proved to be one of the significant difficulties facing humanity. United Nations forecasts estimate that half of the populace in Africa and Asia will reside in the urban and peri-urban regions by the year 2020 [7,8]. According to FAO [10], there has been rapid economic growth associated with rapid rates of urbanisation which is evident in sub-Saharan Africa, including South Africa. Increasing urbanisation coupled with increased poverty parallel to growing populations becomes a challenge if employment prospects remain low. In addition to the identified poverty state, the food insecurity problem is shifting from rural areas to urban centres [11,12].

Furthermore, to be able to tackle the food insecurity problem globally adequately, greater consideration has to be given to access and accessibility of essential services such as clean water, acceptable health care for the poor and sanitation [13,14]. Underprivileged inhabitants in the urban centres of developing countries confront many challenges in securing livelihoods. This situation, in turn, disturbs the food security status of the household and especially those of susceptible groups that include, children, women, the disabled and elderly [11,15]. Males generally participate in the more skilled and physical labour job market; meanwhile women are susceptible to the unskilled labour market [11,12,16]. Due to the skewed distribution of employment opportunities by gender distribution, both female and male-headed households face different difficulties in acquiring basic needs and food. Male-headed households generally are characterised with higher incomes as compared to female-headed households [17]. On the other hand, females are said to be able to discover more inventive methods to earn income and to find food to sustain themselves and maintain their families. A traditional noted method that women engage in is UPA [18,19].

Urban and peri-urban agriculture comprises agricultural activities that occur in intra-urban areas of metropolises and cities and developed peri-urban fringes [20,21]. These activities range from production, processing, distribution and marketing of agricultural products [22,23]. Returns from UPA activities are either food crops or livestock [24–26]. Urban and peri-urban agriculture occupies vacant land usually situated along river banks, roadsides and streams and in wetlands [24,27]. Nowadays, UPA is on the increase in sub-Saharan African cities regardless of some of the challenges of access to basic services and land tenure. According to Statistics South Africa [17], South Africa's urban population has increased, with Gauteng Province having the highest population of 12.2 million people in 2011 followed by Kwa-Zulu Natal with 10.3 million people. This population influx is due to the labour movement of job seekers to the larger metropolises from rural South Africa [13,28].

In South Africa, particularly in Kwa-Zulu Natal, urbanisation was delayed due to influx control laws which were in place during South Africa's apartheid past [29]. Today, rural-urban migration is one of the main factors in the rising levels of slum sprawl in urban and peri-urban areas. Poverty for several years has been linked to rural communities and labelled as a rural phenomenon, therefore, it has been the critical driving force of people migrating to urban areas in hopes to pursue better opportunities. Therefore, it has become a significant challenge to ensure food provision for poor urban citizens. Urban and peri-urban agriculture may be regarded as a potential resolution to the challenge for urban food insecurity for the urban poor as these poor urbanised citizens may be open to practising agriculture due to their rural backgrounds. In other words, feeding a growing urban population living in poverty will be one of the significant humanitarian and political challenges of the next century [29]. Therefore, this suggests that there will be increased pressure for food supply on urban and peri-urban agriculture.

Urban and peri-urban agriculture is noted to be on the increase within small sections in cities, either in vacant plots of land being used to grow food near informal settlements, yards and nearby rivers [30]. These various plots are sustained either by individuals or small groups. The primary purpose is to feed their families and perhaps make additional money to provide for their families and households to be able to sustain their growth potential. Regardless of the use of cutting-edge technologies in agriculture, the existing food systems have not been successful in ensuring food security for the rapidly increasing population worldwide [31]. Viljoen et al. [32] and Ward et al. [33] contend that conventional agriculture cannot guarantee food security for the rapidly increasing population and furthermore, due to the adverse effects that conventional agriculture has on the environment, it is therefore imperative for alternative methods of food production to be explored. Food security needs to undergo policy intervention to improve the implementation of food security strategies within urban and peri-urban spheres [5]. For years, food insecurity has been conventionally theorised to be a common rural developmental challenge. Therefore, the current theoretical tools to comprehend the challenge remains inadequate to address food insecurity in urban areas. Such tools mainly concentrate on issues of accessibility rather than on finding solutions on improving food production within the urban and peri-urban areas.

Although the increase of the practice of urban and peri-urban agriculture in South Africa is becoming common, there are relatively less empirical studies on the implications of the practice of urban and peri-urban agriculture. Again, there are still fewer studies done to indicate the reasons that enable or inhibit the practice of UPA to improve food security by low-income households. There is however some indication that UPA could be enabled or hindered by socio-economic characteristics of households and structural dynamics of power and privilege. The general purpose of this paper, therefore, is to analyse the contribution of urban and peri-urban agriculture towards the food security status of households practising any UPA activities vis-a-vis those households that did not practice any UPA activities in Tongaat peri-urban area of eThekwini Municipality. The specific objectives are to estimate the food security status of the Tongaat peri-urban households and to elicit the reasons that make urban and peri-urban households to practice UPA activities or not. The paper's

underlying assumption is that urban and peri-urban agriculture contributes to the food security status of households both in terms of dietary diversity and access.

## 2. Description of the Study Area

The current study was carried out in Tongaat peri-urban area within the eThekwini Municipality of KwaZulu-Natal Province in South Africa. The selection of Tongaat is due to its productive and potential UPA activities. Tongaat's location is about 37km northbound of Durban [34]. The name Tongaat is synonymous with sugar because this is where Tongaat Hullet Group (agriculture and agro-processing business of sugarcane and maize) has its headquarters and their largest mill. Tongaat is one of the leading sugar producing regions in the world. Tongaat is found between the developmental corridor that exists between Richards Bay and Durban [35]. This area is known for its increasing and urban potential developmental prospects. Tongaat is accessible to the populations living in the surrounding rural areas as it provides convenient transportation. Tongaat makes use of both rail and road to connect the rural communities to Durban city centre [35]. Figure 1 is a map showing the location of Tongaat within the eThekwini Municipality.

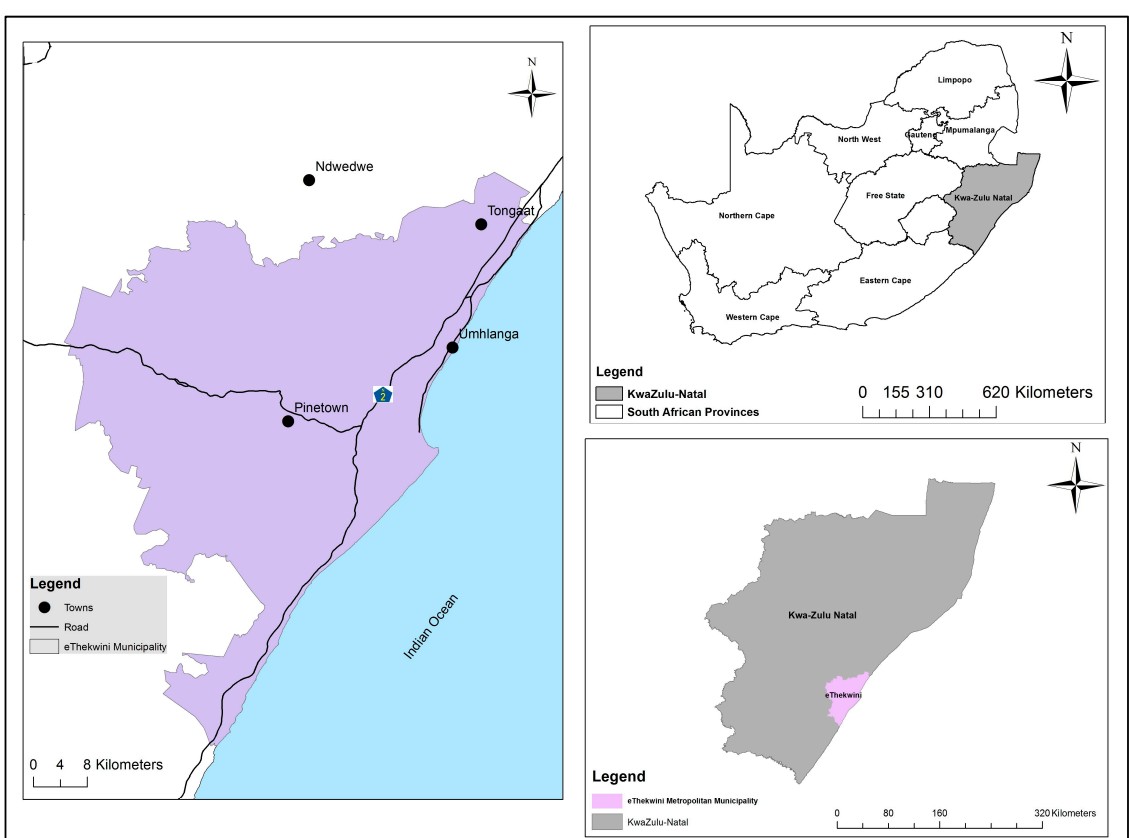

**Figure 1.** Map showing the location of Tongaat. Source: UNIZULU Geography Department [36].

## 3. Methodology

### 3.1. Conceptual Framework for Urban Agriculture as a Tool for Enhancing Household Food Security

The study is grounded in the conceptual framework developed by Kutiwa et al. [37] for urban agriculture as a tool for enhancing household food security. Poor households in urban and peri-urban households are vulnerable to food insecurity due to low incomes, unemployment, poverty and lack of amenities among other factors. Urban and peri-urban agriculture presents an opportunity to escape the poverty and food insecurity cycle by households. Households, as they engage in UPA activities, increase their potential to address the household food security status through adequate access, food

availability and utilisation. Urban and peri-urban agriculture is likely to ensure food availability through the supply of fresh food to a household consistently given that the production resources are available. Additionally, households apart from producing their food for household consumption can also produce for selling and thus generate income. A household with diversified sources of income including income earned from the sale of urban and peri-urban agricultural produce can mobilise resources to access adequate and nutritious food. Along with nutritional security, UPA is also noted to ensure food utilisation, that is the intake of quality food that is suitably used; processed and stored. Figure 2 presents a conceptual framework for urban agriculture as a tool for enhancing household food security.

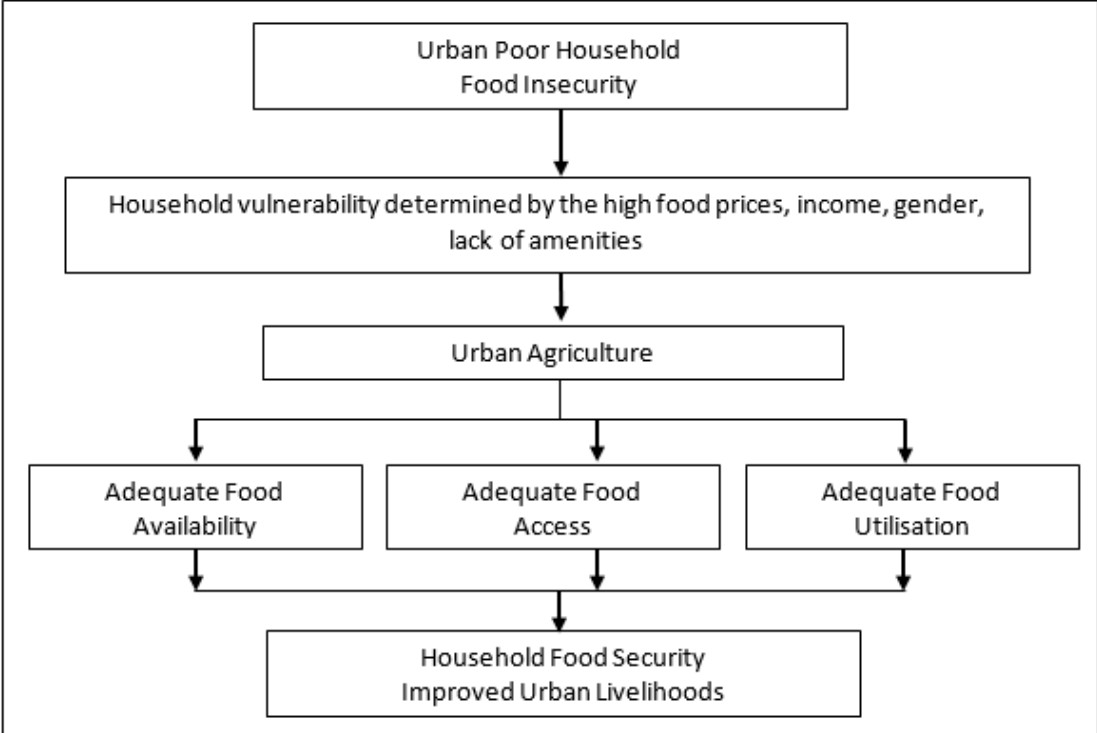

**Figure 2.** A conceptual framework for urban agriculture as a tool for enhancing household food security. Source: Borrowed from Kutiwa et al. [37].

### 3.2. Research Design

The study adopted a quantitative research approach. Quantitative research is important because deductive reasoning moves from general to specific. The results of the research presented in this paper employed a descriptive cross-sectional design in collecting data on relevant variables required from the sample size. This design was suited for this study because it is an inexpensive method and does not require too much time.

### 3.3. Description and Selection of Respondents

This study employed a probability sampling because the targeted sample group was stratified into UPA practising households (that is those that practised any UPA activities) and non-UPA practising households (those that did not practice any form of UPA activities). A selection of a total sample size of 208 respondents (that is 109 and 99 UPA and non-UPA practising households respectively) was through stratified random sampling method. A sample size of 208 respondents was deemed to be large enough to generate meaningful statistical analysis, yet at the same time small enough to be manageable. By making use of the stratified random sampling technique, the researcher intended to highlight differences between specific sub-groups while ensuring greater precision [38]. The study

population of interest in this paper composed of Tongaat peri-urban dwellers (that is both UPA and non-UPA practising households) under the eThekwini Municipality. The actual respondents from the selected households were the individuals that were involved in preparing food for the household or an adult individual in the absence of the former.

*3.4. Data Collection*

Data collection was achieved using a questionnaire through a survey method. Questionnaires administration involved individuals that were involved in preparing the food for the households to answer. As already indicated, the respondents were selected from UPA and non-UPA practising households to allow for a comparative analysis of the dynamics at play concerning the practice of UPA and household food security. Data collection progressed from December 2017 to January 2018. Data collection avoided irregular periods such as funerals, festive events, paydays, weddings and social grant pay-out dates. The rationale behind this is that the study had to reveal the true nature of a household's food security status particularly when a household experiences the greatest food shortages. The questionnaire collected data that included household socio-economic characteristics., for example, Such data included demographics (gender, age, educational level, marital status, employment status, household income); and information relating to the practice of UPA (crops and livestock); HDDS 24-h recall food security information; HFIAS 4-week recall food security information; health-related information (that is whether household members suffered from any food insecurity related diseases for both adults and children). Before the data collection process, six research assistants employed as enumerators underwent training. Respondents answered a translated questionnaire in isiZulu, which is the native language in Tongaat. The questionnaire comprised of both closed and open-ended questions. Before the actual survey, the researcher conducted a pilot test. The test was able to determine the viability of the study before continuing with the major research. The University of Zululand issued an ethical clearance to conduct the study. The administration of the questionnaire to respondents was through face-to-face interviews. The benefits of an interviewer-administered questionnaire are that respondents can seek clarity from the researcher and reduces confusion for the respondent [39].

*3.5. Data Analysis*

After data collection, data management (capturing, coding and cleaning) was carried out in Microsoft Excel 2016 and later exported to Statistical Package for Social Sciences (SPSS) version 25 and STATA software (version 14) for analysis. Descriptive analysis was applied to describe the socio-economic characteristics of respondents and the major UPA activities within the study area. To estimate the household food security status for the interest groups—that is UPA and non-UPA practising households, the HDDS and HFIAS indices were computed. A Pearson Chi-Square test of association was performed to discover if there were significant relationships between the household food security status (proxied by the HDDS—that is food secure when the HDDS fell above the mean and otherwise when the HDDS fell below the mean) and the categorical independent (explanatory) variables). A probit regression analysis was used to elicit the reasons associated with the practice of any UPA activities or not by a household. The sections that follow describe these instruments.

3.5.1. Computing the Household Dietary Diversity Score (HDDS) and the Household Food Insecurity Access Score (HFIAS)

The HDDS is a commonly used instrument to measure household food security. This paper adopts the HDDS tool from Swindle and Blinksy [40]. The food consumption calculation was done using 12 food groups namely "cereals, tubers, vegetables, fruits, meat, eggs, fish, beans, dairy products, fats/oils, sugar/honey and condiments." Respondents were asked to specify the type of foods consumed among the households in a 24-h recall consumed at the household level. An improvement in a household's score reveals that there has been a substantial enhancement in the household's diet. The calculation of the HDDS was for every household that participated in the study. The estimation of

the HDDS was by adding the quantity of the consumed food groups either by the household or by an individual throughout a 24-h recall period. According to Rajendran [41], the HDDS does not have restrictions regarding the number of food groups to indicate adequate or inadequate dietary diversity. Accordingly, the recommendation is that the distribution or average of scores should be in use in order to analyse data as accurate as possible. Households that fell above the mean HDDS were considered to be food secure and those households that fell below the mean HDDS were considered to be food insecure. The following represents the average HDDS calculation (Equation (1)):

$$Average\ HDDS = \frac{Sum\ (HDDS)}{Total\ number\ of\ households} \tag{1}$$

### 3.5.2. Computing the Household Food Insecurity Access Scale (HFIAS)

This paper, in addition to the HDDS analysis, employs the HFIAS tool to supplement the results of the HDDS tool as recommended by other scholars [42–44]. This paper adopts the HFIAS tool from Coates et al. [45]. The HFIAS score represents the degree in which a household found themselves food secure or insecure for the preceding four weeks at the time of the study. The estimation of a household's HFIAS score was by adding the frequency of occurrence codes for each question for each household by adding the codes for each frequency-of-occurrence question. The HFIAS score ranges from 0 to 27 [45]. A household that attained a higher than the average score suggests that it may be food insecure. Similarly, a household that attained a lower than the average score would suggest that it may be food secure. The following depicts how the HFIAS is calculated (Equation (2)):

$$HFIAS\ Score = Sum\ frequency\ of\ occurrence\ question\ response\ codes\ (Q1a + Q2a \ldots + Q9a) \tag{2}$$

The average HFIAS is computed as follows (Equation (3)):

$$Average\ HFIAS\ score = \frac{Sum\ of\ HFIAS\ in\ the\ sample}{Total\ number\ of\ HFIAS\ scores\ in\ the\ sample} \tag{3}$$

The classification of the HFIAS is into four food security groups which the households could fall in namely: "food secure, mildly food secure, moderately food insecure and severely food insecure." Food secure households are those that experience none or are rarely concerned about food insecurity (access) conditions. A household is considered to be mildly food secure when the household sometimes or often worries about not being able to obtain and consume desired foods, not having access to enough food or result in consuming a more monotonous diet than preferred, or, conversely rarely, eat some foods deemed undesirable. A household that is deemed to be mildly food insecure is one that sacrifices quality very often, for example through eating a monotonous diet, or, occasionally undesirable foods. Such a household would at times, nonetheless rarely, decrease on the amount of food consumed (that is the number or size of meals), although the household may not somewhat experience the three extreme severe food insecurity circumstances. A household that is deemed to be severely food insecure would regularly graduate to decrease their meal portions or on the number of meals or experiences any of the three utmost severe circumstances (that is of food shortage, spending the entire day and night without a meal or going to sleep starving).

To determine whether there were significant differences between the food security status in terms of dietary diversity (HDDS) and food access (HFIAS) of the two groups (that is the UPA and Non-UPA households); a non-parametric (Mann-Whitney U) test was applied.

### 3.5.3. The Probit Regression Analysis

Based on the cumulative normal probability distribution, the probit regression analysis takes on the values zero (0) and one (1) and is a commonly used model in social science research to analyse data where the dependent variable is dichotomous. A probit regression model was used to elicit the reasons

that make the urban and peri-urban households to practice UPA activities or not. The probit regression model estimates the probability that observation with particular characteristics will fall into a specific one of the categories; moreover, classifying observations based on their predicted probabilities is a type of binary classification model. In this paper, the dependent variable of interest was whether a household practised any form of UPA activities or otherwise. The outcome of the dependent variable has the probability of belonging to one of the two conditions, which can take on any value between 0 and 1. Households who practised any form of peri-urban agriculture were assigned a numeric code of one (1) and zero (0) for households that did not practice any form of peri-urban agriculture. Modelling the (conditional) probability of a "successful" outcome, that is, $Y_i = 1$ can be expressed as follows (Equation (4)):

$$P\left[Y_i = 1 \mid X_{1i}, \ldots, X_{ki}; \beta_0, \ldots, \beta_k\right] = \Phi(\beta_0 + \sum_{k=1}^{K} \beta_k X_{ki}) \tag{4}$$

Where $\Phi$ is the cumulative distribution function of the standard normal distribution. This means that conditional on the regressors, the probability that the outcome variable, $Y_i = 1$, is a certain function of a linear combination of the regressors. The specification of the linear regression is as follows (Equation (5)):

$$E\left(Y_i \mid X_{1i}, \ldots, X_{ki}; \beta_0, \ldots, \beta_k = \beta_0 + \sum_{k=1}^{K} \beta_k X_{ki} \tag{5}$$

Other than in the linear regression model, coefficients rarely have any direct interpretation. In the probit regression model, the interpretation of the relationship between a specific explanatory variable and the outcome of the probability is through the marginal effects. The marginal effect associated with the explanatory variables accounts for the partial change in the probability ceteris paribus (holding the other variables constant), that is, the effects of changes in the regressors affecting the features of the outcome variable. The marginal effect associated with continuous explanatory variables $X_k$ on the probability $P(Y_i = 1 \mid X)$ can be derived as follows (Equation (6)):

$$\frac{\partial P\left[Y_i = 1 \mid X_{1i}, \ldots, X_{ki}; \beta_0, \ldots, \beta_K\right]}{\partial X_{ki}} = \beta_k \Phi(\beta_0 + \sum_{k=1}^{K} \beta_k X_{ki}) \tag{6}$$

The marginal effect on dummy variables is computed differently from continuous variables. Discrete changes in the predicted probabilities constitute an alternative to the marginal effect when interpreting the influence of a dummy variable. In the case of discrete regressors, the discrete change in a regressor $X_{ki}$ takes the values and is derived as follows (Equation (7)):

$$\begin{aligned}
\Delta X_{ki} \, & P\left[Y_i = 1 \mid X_{1i}, \ldots, X_{ki}; \beta_0, \ldots, \beta_K\right] \\
&= \beta_k \Phi(\beta_0 + \sum_{l=1}^{k-1} \beta_l X_{li}) + \beta_k + + \sum_{l=k+1}^{k-1} \beta_l X_{li}) \\
&- \beta_k \Phi(\beta_0 + \sum_{l=1}^{k-1} \beta_l X_{li} + \sum_{l=k+1}^{k-1} \beta_l X_{li})
\end{aligned} \tag{7}$$

Scientific literature especially in the field of econometrics, illustrate the probit model as follows (Equation (8)):

$$\Pr(Y = 1 \mid X) = \beta_0 + \beta_n X + \varepsilon \tag{8}$$

Before computing the probit regression model, this paper employed a bivariate analysis in order to explore whether a functional relationship exists between the variables or not. There are different types of bivariate analysis such as the scatterplot, regression analysis and the correlation coefficients (depending on whether the variables are numerical, categorical or both) in order to identify the significant variables. This paper makes use of the correlation coefficients. The correlation coefficients indicate whether the variables in question are related. A zero (0) coefficient suggest that the variables

are not correlated (that is, there is no association), while a coefficient of one (1) (either positive (+) or negative (−)) means that the variables are perfectly correlated (that is, they are entirely in sync with each other). For the probit regression model, it incorporated only those variables that were significant from the bivariate analysis.

The selection of the independent (explanatory) variables that were considered likely to be the reason to practice any form of UPA activities by a household relies on literature studies. These include age of the respondent, household size (family members), educational level (measured in schooling years), gender of the respondent (male/female), employment status of the respondent (employed/unemployed), household access to farming inputs/implements (yes/no), household receiving any social grant (yes/no), arable land size (hectares), land tenure (ownership rights of arable land by the household) (yes/no), average monthly household income (US$) and the food security status of households (proxied by the HDDS and HFIAS measures).

## 4. Results

### 4.1. Summary of Socio-economic Characteristics of Respondents

A study by Arene and Anyaeji [46] reveal that households that are led by older members are more likely to be food secure than those headed by younger members. The results of this study suggest that the respondents were at their economically active years and could partake in agricultural activities and as also earn an income. The mean age was about 45 and 47 years old for the UPA and non-UPA practising households respectively. Table 1 shows that the age of the respondents ranged from 21 to 73 years and 22 to 70 years for UPA and non-UPA practising households respectively. The results show that the average age of non-UPA practising households was slightly higher than that of UPA households. The mean household size was about 10 and 9 members for the UPA and non-UPA practising households respectively. Household members consisted of parents, grandparents, grandchildren and other extended family members. It is, therefore, more likely that households with more members rely on UPA produce to keep the household food secure; therefore, constant food availability may motivate them to participate in UPA activities. Altman et al. [47], agree that households with larger groups of dependants encourage the engagement of the household in subsistence production because there is a higher demand for food.

Overall, the results show that the maximum number of schooling years was 19 years. Households that practised any form of UPA activities, however, appear to have a greater (19 years) maximum number of schooling years than their counterparts (non-UPA practising households—15 years). The overall average number of schooling years was 9.45 years for all households; 9.17 years for UPA practising households and 9.16 for non-UPA households. The average number of schooling years between UPA and non-UPA practising households suggests that the education levels between the two groups were more or less the same.

In this paper, it is evident that there were more female respondents that is 56 and 53 percent for the UPA and non-UPA practising households respectively than males. Indeed, a similar observation was noted by Ngome and Foeken [19], that women tend to be the majority in the agricultural sector. The findings show that men tend to have lesser participation in agricultural activities partly because they may be engaged in other activities (jobs) to secure an income for the family (household). However, a Pearson Chi-Square test ($\chi^2$ = 0.181 df = 1, $p$ = 0.671) for all households' analysis shows that there is no statistically significant association in this dataset between the household food security status of households (that is food secure when the HDDS fell above the mean and otherwise when the HDDS fell below the mean) and the gender variable. Single respondents accounted for about 40 and 35 percent for UPA and non-UPA practising households respectively.

**Table 1.** Summary of socio-economic characteristics of respondents.

| Variable | | UPA Practising Households | | | | Non-UPA Practising Households | | | | All Household (Combined Analysis) | | | |
|---|---|---|---|---|---|---|---|---|---|---|---|---|---|
| | | Mean | Min | Max | SD | Mean | Min | Max | SD | Mean | Min | Max | SD |
| Age (years) | | 44.68 | 21 | 73 | 16.053 | 46.84 | 22 | 70 | 12.09 | 45.71 | 21 | 73 | 15.552 |
| Household size | | 9.68 | 2 | 26 | 5.366 | 8.63 | 2 | 22 | 3.760 | 9.16 | 2 | 26 | 4.690 |
| Educational level (years) | | 9.17 | 2 | 19 | 3.441 | 9.18 | 2 | 15 | 3.288 | 9. 17 | 2 | 19 | 3.372 |
| **Variable** | | **Percentage** | | | | **Percentage** | | | | **Percentage** | | | |
| Gender | Female | 56.0 | | | | 53.5 | | | | 54.8 | | | |
| | Male | 44.0 | | | | 46.55 | | | | 45.2 | | | |
| Employment status | Unemployed | 65.1 | | | | 34.3 | | | | 50.5 | | | |
| | Employed | 34.9 | | | | 65.7 | | | | 49.5 | | | |
| Access to farming inputs/implements | No | 89.9 | | | | 80.8 | | | | 85.6 | | | |
| | Yes | 10.1 | | | | 19.2 | | | | 14.4 | | | |
| Receiving social grants | No | 16 | | | | 14.7 | | | | 14 | | | |
| | Yes | 84 | | | | 85.3 | | | | 86 | | | |
| Type of social grant received | Child | 50.4 | | | | 44.4 | | | | 47.6 | | | |
| | Disability | 2.8 | | | | 6.1 | | | | 4.3 | | | |
| | Pension | 15.6 | | | | 15.2 | | | | 15.4 | | | |
| | Child and Pension | 16.5 | | | | 20.2 | | | | 18.3 | | | |
| | No grant received | 14.7 | | | | 14.1 | | | | 14.4 | | | |
| Land size (ha) | <1 | 3.7 | | | | 57.6 | | | | 29.3 | | | |
| | 1–3 | 37.6 | | | | 23.2 | | | | 30.8 | | | |
| | 4–6 | 36.7 | | | | 8.1 | | | | 23.1 | | | |
| | 7–9 | 11.0 | | | | 7.1 | | | | 9.1 | | | |
| | >10 | 11.0 | | | | 4.0 | | | | 7.7 | | | |
| Land tenure | own land | 19.3 | | | | 8.1 | | | | 13.9 | | | |
| | Renting land | 11.9 | | | | 0.9 | | | | 6.8 | | | |
| | Sharing/communal land | 21.1 | | | | 5.1 | | | | 13.5 | | | |
| | Unspecified land | 44.0 | | | | 28.3 | | | | 36.5 | | | |
| | No land | 3.7 | | | | 57.6 | | | | 29.3 | | | |
| Average monthly household income (US$) | 0–36 | 6.4 | | | | 3.0 | | | | 4.8 | | | |
| | 37–72 | 3.7 | | | | 4.0 | | | | 3.8 | | | |
| | 73–108 | 15.6 | | | | 16.2 | | | | 15.4 | | | |
| | 109–144 | 15.6 | | | | 11.1 | | | | 13.5 | | | |
| | 145–180 | 13.8 | | | | 19.2 | | | | 16.3 | | | |
| | >180 | 45.0 | | | | 47.5 | | | | 46.2 | | | |

Min; Max; SD; US$ denotes Minimum; Maximum; Standard Deviation and United States Dollar as at 6 December 2018. Source: Survey data (2017/18).

The results reveal a high unemployment rate with about 65 and 34 percent for the UPA and non-UPA practising households. The high unemployment rate can make households to be extremely vulnerable to poverty and food insecurity. A Pearson Chi-Square test ($\chi^2$ = 6.240 df = 1, $p$ = 0.012) for all households' analysis shows that there is a statistically significant association between the household food security status of a household (that is food secure when the HDDS fell above the mean and otherwise when the HDDS fell below the mean) and employment status. The finding shows that a majority (86%) of urban and peri-urban households relied largely on social grants with those that received the child support grant accounting for a higher (about 51% and 44%) proportion for the UPA and non-UPA practising households. Households that did not obtain any form of government social grant constituted about 15 and 14 percent of the UPA and non-UPA practising households respectively. However, a Pearson Chi-Square test ($\chi^2$ = 1.863 df = 1, $p$ = 0.172) for all households' analysis shows that there is no statistically significant association in this dataset between the household food security status of households (that is food secure when the HDDS fell above the mean and otherwise when the HDDS fell below the mean) and whether the household received any social grant.

Access to farming inputs and implements is also of crucial importance in UPA activities to stimulate production for example access to composted or fresh organic material and other sources of nutrients (like wastewater) and efficient water irrigation systems (such as drip irrigation) among other implements. The results show that the majority (about 86%) of urban and peri-urban households (all households) did not have adequate access to farming inputs and implements. However, a Pearson Chi-Square test ($\chi^2$ = 1.029 df = 1, $p$ = 0.310) for all households' analysis shows that there is no statistically significant association in this dataset between the household food security status of households (that is food secure when the HDDS fell above the mean and otherwise when the HDDS fell below the mean) and whether the household had access to farming inputs and implements.

Access to arable land is a crucial element in ensuring food security. The results show that a higher proportion (about 38%) of the UPA practising households had access to an arable land size ranging between 1 to 3 ha. Land ranging between 7 to 9 ha and >10 ha were each owned by 11% of the UPA practising households. About 4 percent of the UPA practising households did not have any access to arable land that they owned; rather they practised their UPA activities on small vacant spaces outside their homes and backyards. Concerning non-UPA practising households, the majority (about 58%) had limited access to arable land with <1 ha of land available to them. About 23 percent of the non-UPA practising households had access to an arable land size ranging from 1 to 3 ha. The minority (about 14%) of non-UPA practising households were made up of the households that had access to >10 ha of land. The non-UPA practising households who had access to land but did not utilise it themselves either left the land fallow or leased it to other UPA practising households. A Pearson Chi-Square test ($\chi^2$ = 9.283 df = 4, $p$ = 0.054) for all households' analysis shows that there is a statistically significant association between the household food security status of a household (that is food secure when the HDDS fell above the mean and otherwise when the HDDS fell below the mean) and the land size variable.

Concerning land tenure, the results show that a higher proportion (44%) of the UPA practising households practised their UPA activities on unspecified land. Only a small proportion (about 19%) of the UPA practising households had land ownership rights (owned the land they farmed). Those who practised sharecropping (that is the households that depended on communal land) accounted for about 21 percent of the UPA practising households. A Pearson Chi-Square test ($\chi^2$ = 3.221 df = 1, $p$ = 0.0730) for all households' analysis shows that there is a statistically significant association between the household food security status of households (that is food secure when the HDDS fell above the mean and otherwise when the HDDS fell below the mean) and whether the household had arable land ownership rights.

An income that is available to a household determines what they can afford and the quantity in which they can afford to buy [48]. A higher income level is advantageous to households as they can afford more and have a greater variety to choose. The majority (55 and 52%) of the UPA and

non-UPA practising households respectively indicated that they earned an average monthly income of lesser than US$180. The minimum stipulated wage in South Africa is US$252 for 40 h and US$281 for 45 h [49]. A Pearson Chi-Square test ($\chi^2$ = 23.133 df = 6, *p* = 0.001) for all households' analysis shows that there is a statistically significant association between the household food security status of a household (that is food secure when the HDDS fell above the mean and otherwise when the HDDS fell below the mean) and the average monthly household income variable.

### 4.2. Urban and Peri-urban Agricultural Activities within Tongaat

Table 2 shows the various types of agricultural practices performed by interviewed households in Tongaat peri-urban areas. A higher proportion (about 46%) of the interviewed UPA practising households in Tongaat were involved with field crops (which included cabbages, spinach, sweet potato and avocado) followed by livestock rearing which accounted for about 40 percent of the sample, poultry (about 35%), fruits (about 4%) and flowers (about 3%). Livestock (both small and large stock) is often preferred to poultry farming because it generates more substantial income and also kept for social status. The area is predominately a sugar farming region. Aside from sugar farming, numerous farms are into crop and livestock production. The UPA practising households predominantly practised field crops because they provide households with staple food items for their diets.

**Table 2.** Urban and peri-urban agricultural activities within Tongaat.

| Type of Agricultural Practice | UPA Practising Households |
|---|---|
| | Percentage (%) |
| Field crops | 45.9 |
| Poultry | 34.9 |
| Flowers | 2.9 |
| Fruits | 3.7 |
| Livestock (both small and large stock) | 40.4 |

Source: Survey data (2017/18).

### 4.3. Source of Income by Tongaat Households

Table 3 shows the sources of income for the interviewed households in Tongaat. The majority (about 86%) of the total sample specified that they were recipients of the government social grant (that is about 85 and 86 percent for the UPA and non-UPA practising households respectively). Paid jobs were reported as a source of income by about 38 percent of the total sample (that is about 38 and 40 percent of the UPA and non-UPA practising households respectively). The minority (about 32%) of the total sample was made up of the respondents that were self-employed (that is about 26 and 38 percent of the UPA and non-UPA practising households respectively). These respondents were self-employed for example in hairdressing and as carpenters. Qualitative data revealed that the majority of the households did not derive income from their UPA activities as they produced mostly for home consumption. The motivation here for a greater share of own household consumption rather than for commercial reasons was the types of crops produced (refer to Section 4.2) (in smaller quantities) and thus dedicated for only household fresh consumption.

**Table 3.** Source of income by Tongaat households.

| Sources of Income | UPA Practising Households | Non-UPA Practising Households | All Households (Combined Analysis) |
|---|---|---|---|
| | Percentage (%) | Percentage (%) | Percentage (%) |
| Paid job | 37.6 | 39.4 | 38.4 |
| Social grant | 85.3 | 85.6 | 85.6 |
| Self-employment | 25.7 | 38.4 | 31.7 |

Source: Survey data (2017/18).

### 4.4. Types of Food Consumed by Tongaat Households

As already indicated, the food consumption calculation was done using 12 food groups. Respondents here indicated the type of foods consumed among the households in a 24-h recall (Table 4). Table 4 shows that more (about 81%) household ate cereals (millet, sorghum, maize and wheat) (that is about 86 and 75 percent for UPA and non-UPA practising households respectively). Household respondents that ate foods made from roots and tuber constituted about 78 percent of the total sample (that is about 88 and 68 percent of the UPA and non-UPA practising households respectively). The least consumed food type was the dairy product with a minority (about 46%) of the total sample reported that they consumed dairy products (that is about 47 and 46 percent of the UPA and non-UPA practising households respectively). Cereals were the most commonly consumed main ingredient since maize meal that is used for preparing pap and porridge is a popular cultural staple food in the study area. The second most important component of the diet for the interviewed households were roots and tubers. Starch is a type of carbohydrate which forms part of the total carbohydrates, including dietary fibre and sugars [50]. The consumption of starch has the potential of raising blood sugar and contributes to the number of calories consumed. It is essential to monitor the consumption of starchy foods because it has the same effects as eating sugary foods. Consuming starchy foods with lack of exercise contributes to raising blood sugar levels and weight gain contributing to diet-related problems such as obesity.

**Table 4.** Types of food consumed by Tongaat households.

| Food Types Consumed | UPA Practising Households | Non-UPA Practising Households | All Households (Combined Analysis) |
|---|---|---|---|
| | Percentage (%) | Percentage (%) | Percentage (%) |
| Any bread, mabele, rice, noodles, biscuits, scones, fatcakes, other food made from millet, sorghum, maize, wheat? | 86.2 | 74.7 | 80.8 |
| Any potatoes and sweet potatoes or any foods made from roots and tubers? | 88.1 | 67.7 | 78.4 |
| Any yellow or orange and green vegetables? | 78.0 | 66.7 | 72.6 |
| Any fruits? | 78.9 | 68.7 | 74.0 |
| Any beef, pork, lamb, mutton, chicken or other birds, liver, kidney, hearts and other organ meats? | 69.7 | 69.7 | 69.7 |
| Any eggs? | 70.6 | 66.7 | 70.2 |
| Any fresh fish or dried fish? | 67.0 | 59.6 | 63.4 |
| Any foods made from beans, peas or lentils? | 66.1 | 61.6 | 65.4 |
| Any dairy products: milk, yoghurt, cheese? | 46.8 | 45.5 | 46.2 |
| Any foods contain fat, butter or oil? | 68.8 | 68.7 | 68.8 |
| Any sugar or honey? | 70.6 | 71.7 | 71.2 |
| Condiments: tea, coffee, sauces, cold drink, juice? | 59.6 | 63.6 | 61.5 |

Source: Survey data (2017/18).

### 4.5. Household Dietary Diversity Score According to the Three Classes/Groups Consumed by Households in Tongaat

Overall, the average mean HDDS score was about 5 (all households). Results from both groups (the UPA and non-UPA practising households) show that close to half (about 48%) of the households were above the mean HDDS level (Table 5). Using the mean HDDS as a cut-off point, households which fell below the mean HDDS level can be regarded as food insecure and that fell above the mean HDDS level can be regarded as food secure. Therefore, overall, the interviewed households could be regarded as moderately food secure. However, it is imperative to note that a larger proportion (about 54%) of the UPA practising households who were above the mean HDDS score was higher

when compared to the non-UPA practising households (about 40%) who fell above the mean HDDS. The explanation here is that UPA practising households do not rely much on purchasing food; hence they grow their food without making use of monetary resources [51]. Indeed, FAO [52], reveal that households in developing countries benefit from gardening activities which act as a primary source of food. The gardening activities enable households to meet their household consumption requirements. The results of this study support this. However, a Mann-Whitney U test (U = 5292, $p$ = 0.808) show that there was no significant difference from this dataset in terms of the dietary diversity (HDDS) of the two groups and we fail to reject the null hypothesis that there is no significant difference (Table 5).

**Table 5.** Household dietary diversity score (HDDS) according to the three classes/groups consumed by households in Tongaat.

| HDDS Level | UPA Practising Households | Non-UPA Practising Households | All Households (Combined Analysis) |
|---|---|---|---|
| | Percentage (%) | Percentage (%) | Percentage (%) |
| <3 (low dietary diversity) | 11.9 | 12.1 | 12.0 |
| 4–5 (medium dietary diversity) | 33.9 | 47.5 | 40.4 |
| >6 (high dietary diversity) | 54.2 | 40.4 | 47.6 |
| Total | 100.0 | 100.0 | 100.0 |
| Mean HDDS score | 5 | 4.5 | 4.75 |

U = 5292, $p$ = 0.808. Source: Survey data (2017/18).

### 4.6. Food Security Status and Health-Related Issues within Households in Tongaat

The food security status of a household may affect the health status of household members. Several studies have revealed linkages between adverse health outcomes and food insecurity among children [53–55]. Studies relating to the health effects of food insecurity in adults tend to be scarce because studies generally concentrate on the relationship between food insecurity and self-reported diseases. Respondents of the interviewed households were requested to indicate the occurrence of some food insecurity related diseases among its members (for both adults and children) (that is if they had any household member/s that suffered from any of the indicated food insecurity related diseases at the time of the study). The various food insecurity related health issues (diseases) used in this paper (Table 6) were guided by a study by Bhawra et al. [16].

Results show that obesity was the common food insecurity related disease among adults as reported by almost half (49%) of the total sample. Obesity, however, appeared to be a more reported food insecurity related issue among the non-UPA practising households (as indicated by the majority (about 63%) of this strata) as compared to about 39 percent of their counterparts (the UPA practising households) who reported the same food insecurity related issue among its adult members. Diabetic adults constituted about 40 percent of the total sample (that is about 41 and 40 percent of the UPA and non-UPA practising households respectively). Other food insecurity related diseases reported by the interviewed households suffered by the adult household members for the total sample included hypertension (30.8%); hyperlipidaemia (21.1%); diarrhoea (25.0%); osteoporosis (13.9%); iron deficiency anaemia (13.9%) with adults that suffered from heart attacks constituting the minority (0.4%).

Results also show that obesity was also reported as a food insecurity related issue among children household members of the interviewed households in Tongaat. Overall, about 43% of the total sample reported obese children in their households. This problem appeared to be more prevalent among the non-UPA practising households (as reported by the majority (about 68%) of this strata as compared to about 19 percent of their counterparts (the UPA practising households) who reported the same food insecurity disease among its children household members. Children in the household that suffered from diarrhoea constituted about 13 percent of the total sample (that is about 8 and 17 percent of the UPA and non-UPA practising households respectively). Other food insecurity related diseases

reported by the interviewed households suffered by children household members for the total sample included malnutrition (8.17%); underweight (7.2%); rickets (6.7%) and with no children that were reported to suffer from Kwashiorkor.

A study by Seligman et al. [56] reveals that there is a relationship between food insecurity and clinical evidence of diet-sensitive chronic diseases. This supports the finding of this study because overall, non-UPA practising households reported to frequently have diet sensitive diseases among its household members (for both adults and children) as compared to the UPA practising households. As revealed by the results of this paper, the households of Tongaat ate starchy based foods which affect their blood sugar levels and could lead to weight gain. Obesity in South Africa is a huge problem and it is evident in the results of this study. Child obesity is also problematic as it sets them up for serious health problems later on in life. About 13 percent of children are overweight in South Africa which is more than double the global average of 5 percent [57]. Again, results earlier on revealed that non-UPA practising households had inadequate dietary and access to food which could account or contribute for the reported food insecurity related issues (diseases).

**Table 6.** Food insecurity related diseases affecting household members (both adults and children) of the interviewed households in Tongaat at the time of the study.

| Adults | | | | | | |
|---|---|---|---|---|---|---|
| **Food Insecurity Related Disease** | **UPA Practising Households** | | **Non-UPA Practising Households** | | **All Households (Combined Analysis)** | |
| | **Frequency** | **Percentage (%)** | **Frequency** | **Percentage (%)** | **Frequency** | **Percentage (%)** |
| Hypertension | 34 | 31.2 | 30 | 30 | 64 | 30.8 |
| Hyperlipidaemia | 35 | 32 | 9 | 9.1 | 44 | 21.2 |
| Diabetes | 45 | 41.2 | 40 | 40 | 84 | 40.4 |
| Obesity | 39 | 35.7 | 62 | 62.6 | 101 | 48.6 |
| Heart attacks | 1 | 0.91 | 0 | 0 | 1 | 0.4 |
| Diarrhoea | 28 | 25.7 | 24 | 24 | 52 | 25 |
| Osteoporosis | 13 | 11.9 | 16 | 16.2 | 29 | 13.9 |
| Iron deficiency anaemia | 21 | 19.2 | 8 | 8 | 29 | 13.9 |
| Children | | | | | | |
| **Food Insecurity Related Disease** | **UPA Practising Households** | | **Non-UPA Practising Households** | | **All Households (Combined Analysis)** | |
| | **Frequency** | **Percentage (%)** | **Frequency** | **Percentage (%)** | **Frequency** | **Percentage (%)** |
| Malnutrition | 14 | 12.8 | 3 | 3 | 17 | 8.17 |
| Obesity | 21 | 19.3 | 68 | 68 | 89 | 42.8 |
| Underweight | 11 | 10 | 4 | 17.2 | 15 | 7.2 |
| Rickets | 14 | 12.8 | 0 | 0 | 14 | 6.7 |
| Diarrhoea | 9 | 8.3 | 17 | 17.2 | 26 | 12.5 |
| Kwashiorkor | — | — | — | — | — | — |
| Iron deficiency anaemia | 13 | 11.9 | 9 | 9.1 | 22 | 10.6 |

Source: Survey data (2017/18).

*4.7. Household Food Insecurity Access Scale (HFIAS) Categories*

The HFIAS category values were calculated for each household by assigning a code for the food insecurity category in which it falls. There are four food security categories which the households could fall under namely: food secure, mildly food secure, moderately food insecure and severely food insecure (Section 3.5.2). Concerning food access by the interviewed households, overall, results show that the majority (about 66%) of the total sample specified that they did not or rarely worried about food shortages (deemed to be food secure) (Table 7). However, a higher (about 72%) proportion of the UPA practising households did not or rarely worry about food shortages when compared to their counterparts, the non-UPA practising households (about 61%) in the same HFIAS category. Those households that indicated that they occasionally or frequently worry about not having sufficient food (categorised as mildly food secure) constituted about 14 percent of the total sample (that is about 15 and 14 percent of the UPA and non-UPA practising respectively). The minority (about 7%) of

the total sample was made up of the household that frequently cut down on the number of meals or their meal size (deemed to be severely food insecure). A higher (7%) proportion of this group belonged to the non-UPA practising households as compared to about 4 percent of the UPA practising households who were in this same HFIAS category. It is quite evident from the results in Table 6 that UPA practising households were better off in terms of food access than their counterparts, the non-UPA practising households. The finding is supported by the studies of Bhawra et al. [16] and Shisanya and Hendriks [58] which revealed that households involved in own food production were better off concerning food access than those households that mainly relied only on food purchasing. This is so because households growing their food do not have to rely on financial capital to access food [52]. Further to substantiate this finding, the Mann-Whitney U test (U = 4118.5, *p* = 0.001) show that there was a significant difference in terms of food access (HFIAS) of the two groups (Table 7).

**Table 7.** Household Food Insecurity Access Scale (HFIAS) categories of the interviewed households in Tongaat.

| HFIAS Category | UPA Practising Households | Non-UPA Practising Households | All Households (Combined Analysis) |
|---|---|---|---|
| | Percentage (%) | Percentage (%) | Percentage (%) |
| Food secure (does not or rarely worries about food shortages) | 71.6 | 60.6 | 66.3 |
| Mildly food secure (sometimes or often worries about not having enough food) | 14.7 | 14.2 | 14.4 |
| Moderately food insecure (sacrifice quality more frequently) | 10.0 | 15.1 | 12.6 |
| Severely food insecure (cutting down on meal size or the number of meals) | 3.7 | 10.1 | 6.7 |
| Total | 100.0 | 100.0 | 100.0 |

U = 4118.5, *p* = 0.001. Source: Survey data (2017/18).

### 4.8. What Are the Reasons for Practising or Not Practising UPA Activities by Tongaat Households?

Twelve (12) explanatory variables were included in the bivariate model guided by literature on the possible factors that could influence the practice of UPA activities or not by households. The results from the bivariate model (Table 8) show that only 2 out of the 12 variables were significant. These variables are arable land sizes and HFIAS. Pearson's correlation coefficients with *p* < 0.05 were taken as being significant. These significant variables from the bivariate analysis were further considered in the probit regression model. Further explanations of the statistical significance variables are given in the next sections of the probit regression analysis.

The results of the probit regression model to elicit the reasons associated with the practice of any UPA activity or not by households in Tongaat is presented in Table 9. From the 2 independent variables that were inputted in the model, only the arable land size variable was found to be statistically significant. A model fit test was performed to assess the robustness of the model. The Likelihood Ratio Chi-Square statistics was used to test the goodness-of-fit or predictive efficiency of the model. The log likelihood of the fitted model that is the Likelihood Ratio Chi-Square test is used to ascertain whether all predictors' regression coefficients in the model are simultaneously zero. In this case, the Likelihood Ratio Chi-Square test was significant at *p* < 0.01 indicating a good predictive capacity of the model suggesting this was a good fit model.

The variable arable land size was found to be positively correlated with the practice of any UPA activity by households in Tongaat at 1% significance level with a *p*-value of 0.000. Having limited access to land of <1 ha was used as the base (reference category) and the results show that the practice of any

UPA activity by households increases if a household belonged to larger arable land sizes categories (that is 1–3 ha; 4–6 ha; 7–9 ha and >10 ha) while holding all other things constant.

**Table 8.** Bivariate model results showing associations between the practice of any UPA activity or not by households and the independent (explanatory) variables (n = 208).

| Variable | Pearson Correlation (r) | Sig. (2-tailed) |
| --- | --- | --- |
| Age of the respondent (years) | −0.070 | 0.318 |
| Household size (number of family members) | 0.112 | 0.106 |
| Educational level of the respondent (schooling years) | 0.081 | 0.245 |
| Gender of the respondent (male/female) | 0.024 | 0.727 |
| Employment status of the respondent (employed/unemployed) | 0.012 | 0.864 |
| Access to farming inputs/implements by a household (yes/no) | −0.129 | 0.063 |
| Household receiving any social grant (yes/no) | −0.008 | 0.913 |
| Arable land size (hectares) | 0.497 *** | 0.000 |
| Land tenure (arable land ownership rights) (yes/no) | 0.001 | 0.989 |
| Average monthly household income (US$) | −0.070 | 0.313 |
| Household dietary diversity score (HDDS) | 0.016 | 0.817 |
| Household Food Insecurity Access Scale (HFIAS) | 0.151 ** | 0.037 |

***, ** Denotes correlation is significant at the 0.01 and 0.05 levels (2-tailed). Source: Survey data (2017/18) computed from SPSS software (version 25).

**Table 9.** Probit regression model results on the reasons associated with the practice of any UPA activity or not by households in Tongaat.

| Variable | | Coef. | Std. Err. | z | P>|z| | [95% Conf. Interval] | |
| --- | --- | --- | --- | --- | --- | --- | --- |
| Constant | | −2.167112 | 0.4034307 | −5.37 | 0.000 | −2.957822 | −1.376403 |
| Land size (ha) | <1 | | | | | | |
| | 1–3 | 2.534459 *** | 0.428722 | 5.91 | 0.000 | 1.69418 | 3.374739 |
| | 4–6 | 3.075966 *** | 0.4525778 | 6.80 | 0.000 | 2.18893 | 3.963002 |
| | 7–9 | 2.617203 *** | 0.498762 | 5.25 | 0.000 | 1.639647 | 3.594759 |
| | >10 | 2.804316 *** | 0.5228445 | 5.36 | 0.000 | 1.77956 | 3.829072 |
| Household Food Insecurity Access Scale (HFIAS) | | 0.0270543 | 0.0422361 | 0.64 | 0.522 | −0.055727 | 0.1098355 |
| Number of observations | | | | 208 | | | |
| Log-likelihood | | | | −88.956464 | | | |
| LR chi2 (5) | | | | 109.96 | | | |
| Prob > chi2 | | | | 0.0000 | | | |
| Pseudo R2 | | | | 0.3820 | | | |

*** Denotes statistical significance at the 1% level. Dependent variable: Practising any form of UPA activity (yes (1)/no (0)). Source: Survey data (2017/18) computed from STATA software (version 14).

## 5. Discussion

Generally, across all households, results show that females were the dominant gender in the study area and respondents were in the economically active age group. Overall, respondents were fairly educated. However, formal employment prospects is still a challenge in Tongaat. The results reveal that the unemployment rate in the study area is high. A greater proportion of households depend on government social grants as a source of income in Tongaat. Those that earned income, results indicate that the majority of the interviewed households received a total household income which is below the South African minimum wage rate. Overall, the socio-economic characteristics of the interviewed households indicate a weak livelihood portfolio for the households and suggest that the households are more likely to be susceptible to food insecurity. A Pearson Chi-Square test reveals that only the employment status, access to arable land, land tenure (ownership rights or arable land) and monthly household income were statistically significantly associated with the food security status of households.

Like all the other small-scale farmers elsewhere in South Africa, urban and peri-urban households in Tongaat encountered a challenge of lack of farming inputs and implements for their UPA activities.

Although it could be arguable that some UPA activities such as backyard gardens could require less economic resources, the lack of farming inputs and implements is likely to impede the practice of UPA activities by households and thus compromise the food security status of the households through reduced quantity and quality of food produced by UPA practising households. Nonetheless, a bivariate analysis reveals that this variable was not statistically significantly associated with the practice of any UPA activity by Tongaat households.

The results also suggest that the households of Tongaat were restricted mainly by the scarcity of land in which they have available to practice their UPA activities. However, UPA practising households had more access to land at their disposal as compared to the non-UPA practising households. The scarcity of land could be one contributing factor that may discourage non-UPA practising households from practising any form of UPA activity. The scarcity of land results in households practising UPA activities in small plots of land which can contribute to the decreased quantity of food produced by households and thus rendering them vulnerable to household food insecurity.

Indeed, it is evident from the probit regression model analysis on the reasons associated with the practice of any UPA activity or not by households in Tongaat that access to more arable land by households will promote the practice of UPA activities. Overall, the model predicts that those households with access to larger pieces of arable land are more likely to practice any UPA activities. This finding is in agreement with the descriptive results where lack of access to arable land is a limiting factor on the practice of any UPA activities with many households forced to practice their UPA activities on small plots of land and in some instances on illegal sites. The finding is in line with Bruinsma [59] who revealed that the expansion of arable agricultural land area might significantly increase crop production. Nonetheless, Smith [60] provides a contrasting view that sustainable agricultural intensification should be employed to promote agricultural production instead of expanding agricultural land as there is no more room to expand. This assertion is especially true within the peri-urban areas, where land may be reserved for urbanisation and not for agricultural production.

Generally, the results show that a smaller proportion of households had full control and land rights of the land they practised their UPA activities. Most UPA practising households depended on sharing (communal) or unspecified land which can be very tricky. Land tenure in communal areas has become somewhat controversial because there is an issue of who controls and regulates the land. Under such circumstances, UPA activities are seen to be as an economically inefficient use of property land by authorities (governments and municipalities). This negative view of UPA activities could be so because authorities are usually under the impression that if the land is not economically rented and managed, then it can be inefficiently used. Again, UPA is linked mainly with the image of rural surroundings than of an urban setting. As such, city planners are inclined to be against the practice of UPA because of the desired aesthetics, they would want to preserve. Again, land cannot be accessed equally by everyone whereby some vulnerable groups, for example, unmarried women can be excluded from being allocated land. Again, permanent cropping may not be allowed, authorities or new leaders in time can revoke land. Nonetheless, a Pearson Chi-Square test and bivariate analyses reveal that this variable was not statistically significantly associated with the household food security status nor the practice of any UPA activity by Tongaat households respectively.

Despite the land scarcity, a higher proportion of the interviewed UPA households in Tongaat produced with field crops followed by livestock rearing, poultry, fruits and flowers. Field crops accounted for a higher proportion which provided households with staple items for their diets. The majority of the total sample indicated that they were recipients of the government social grant as their source of income. High reliance on government social grants suggests that households are susceptible to food shortages and food insecurity because they do not have secondary sources of income to provide food for the household. The HDDS and HFIAS analyses reveal that starch (cereals—millet, sorghum, maize and wheat) was the most consumed type of food as indicated by the majority of households. Using the mean HDDS as a yardstick for the food security status of households, the

average mean HDDS was about 5 for the total sample. This finding suggests that households were moderately food secure. However, the majority of households that attained a mean HDDS that was above the mean were noted to belong to the UPA practising households when compared to the non-UPA practising households. However, further analysis employing a non-parametric Mann-Whitney U test show that there was no significant difference from this data in terms of the dietary diversity (HDDS) of the two groups. This paper, therefore, fails to reject the null hypothesis that there is no significant difference between the dietary diversity of the UPA and non-UPA practising households as would have been expected. As evidenced by the descriptive analysis – all household analysis show that all households mostly relied on starchy foods with less food diversification. Again, descriptive results suggest that UPA activities were not the primary source of income for households and thus food procurement. Because urban and peri-urban households largely depend on buying food rather than own food production, therefore their food security depends mostly on whether the household has adequate food purchasing power. Given that most of the Tongaat households were unemployed and having limited livelihood diversification, it made it difficult for the households to be truly food secure (in terms of dietary diversity) from limited own food production from UPA activities. However, from literature UPA practising households are expected to diversify their foodstuffs from UPA activities ranging from home gardens such as growing own vegetables and poultry. As such, it is inconclusive from this paper to conclude that UPA practising households were better off than their counterparts in terms of meeting their dietary diversity requirements.

When people live under limited dietary diversity or if they are forced to use severe coping strategies where they compromise nutrition, this may result in nutrient deficiency which will make them prone to a variety of food insecurity related diseases. Therefore, this makes life more difficult for households because such a situation has a potential not only to decrease the labour productivity of a household but also to increase their health care bills. With regard, to the food insecurity related diseases (issues), obesity was the largest reported health problem suffered by both adults and children for all the households. However, the health-related issue of obesity was frequently observed among the non-UPA practising households, suggesting that the practice of UPA can be beneficial in terms of the health status of households. Concerning the HFIAS analysis, the results confirm that a larger proportion of the UPA practising households had comparatively better access to food than the non-UPA practising households. It is quite evident from these results that UPA practising households were better off in terms of food access than their counterparts, the non-UPA practising households. The results confirm the findings from the literature that UPA can improve the food security status of households in terms of supply (food availability). Further analysis, the non-parametric Mann-Whitney U test shows that there was a significant difference in terms of food access (HFIAS) of the two groups (the UPA and non-UPA practising households).

## 6. Conclusions

The practice of UPA is in line with the UN's 2030 agenda for Sustainable Development Goals (SDGs), previously the MDGs to provide food and ensure food security for households especially in urban and peri-urban areas. In as much as UPA is theorised to be a tool to improve food security among low-income urban households, there is understanding in terms of its contribution on reducing food insecurity and the reasons associated with its practice. In this paper, we assessed the contribution of UPA to household food security in terms of dietary diversity and access by estimating the food security status of UPA practising households vis-à-vis the non-UPA practising households. Overall, all households' analysis shows that a greater proportion of the low-income households were not formally employed and relied on government social grants for disposable income and as a livelihood source. The majority of the interviewed households earned less than (<US$180) which is below the South African minimum wage rate. This finding implies that the majority of the Tongaat urban and peri-urban households were susceptible to food insecurity and could benefit from the practice of UPA. An analysis of the food security status by households reveals that all interviewed households

in Tongaat were moderately food secure in terms of dietary diversity. Although UPA practising households were found to be comparatively better off in terms food access than their counterparts, the non-UPA households, this paper was not able to provide evidence that UPA activities can improve the dietary food diversity by households. An analysis of the prevalent food security related diseases shows that obesity was the most reported food security related issue for both adults and children from the total sampled households. However, the problem of obesity was observed to be prevalent in the non-UPA practising households. It is therefore important to note that households do not only acquire food (access) but make sure that the food they eat is nutritious (of diverse nutrients—balanced diet). The findings from this paper have significant policy implications to note. Own food production by households through UPA activities alone without support may not be an effective tool to improve food security (particularly concerning dietary diversity). Rather a more integrated approach from the government and policy makers (city planners) in terms of integrating UPA into city development plans such as policies for allotment of land for garden spaces. It is also evident from the descriptive statistics that access to arable land was a major challenge hindering the practice of UPA. Furthermore, the probit regression analysis elicits that access to more arable land will encourage the practice of UPA activities by households and thus this may, in turn, improve the food security status of households. A Person Chi-Square test reveals that land size was statistically significantly associated with the food security status of households. There is also a need by households to diversify the UPA activities as most households were predominantly engaged in crop production. Urban and peri-agricultural activities diversification can promote the production of high-value foods for example mushrooms and poultry that may not require large space for production but rather have good monetary returns (income generation), consistent supply of food, improved food nutrition and health. In light of this, the practice of integrated UPA that is supported by government and municipal authorities for example in terms of agricultural land allotment can be taken advantage of by households as it would create employment opportunities and income generation which will in turn also contribute to household food purchasing power thus improve the food security status of households.

The practice of UPA is in line with the UN's 2030 agenda for Sustainable Development Goals (SDGs), previously the MDGs to provide food and ensure food security for households especially in urban and peri-urban areas. In as much as UPA is theorised to be a tool to improve food security among low-income urban households, there is understanding in terms of its contribution on reducing food insecurity and the reasons associated with its practice. In this paper, we assessed the contribution of UPA to household food security in terms of dietary diversity and access by estimating the food security status of UPA practising households vis-à-vis the non-UPA practising households. Overall, all households' analysis shows that a greater proportion of the low-income households were not formally employed and relied on government social grants for disposable income and as a livelihood source. The majority of the interviewed households earned less than (<US$180) which is below the South African minimum wage rate. This finding implies that the majority of the Tongaat urban and peri-urban households were susceptible to food insecurity and could benefit from the practice of UPA. An analysis of the food security status by households reveals that all interviewed households in Tongaat were moderately food secure in terms of dietary diversity. Although UPA practising households were found to be comparatively better off in terms food access than their counterparts, the non-UPA households, this paper was not able to provide evidence that UPA activities can improve the dietary food diversity by households. An analysis of the prevalent food security related diseases shows that obesity was the most reported food security related issue for both adults and children from the total sampled households. However, the problem of obesity was observed to be prevalent in the non-UPA practising households. It is therefore important to note that households do not only acquire food (access) but make sure that the food they eat is nutritious (of diverse nutrients—balanced diet). The findings from this paper have significant policy implications to note. Own food production by households through UPA activities alone without support may not be an effective tool to improve food security (particularly concerning dietary diversity). Rather a more integrated approach from

the government and policy makers (city planners) in terms of integrating UPA into city development plans such as policies for allotment of land for garden spaces. It is also evident from the descriptive statistics that access to arable land was a major challenge hindering the practice of UPA. Furthermore, the probit regression analysis elicits that access to more arable land will encourage the practice of UPA activities by households and thus this may, in turn, improve the food security status of households. A Person Chi-Square test reveals that land size was statistically significantly associated with the food security status of households. There is also a need by households to diversify the UPA activities as most households were predominantly engaged in crop production. Urban and peri-agricultural activities diversification can promote the production of high-value foods for example mushrooms and poultry that may not require large space for production but rather have good monetary returns (income generation), consistent supply of food, improved food nutrition and health. In light of this, the practice of integrated UPA that is supported by government and municipal authorities for example in terms of agricultural land allotment can be taken advantage of by households as it would create employment opportunities and income generation which will in turn also contribute to household food purchasing power thus improve the food security status of households.

**Author Contributions:** N.Z.K. developed the study concept under the supervision of M.S. N.Z.K. did data collection and the compilation of the draft manuscript. Data analysis, scientific validation and scrutiny were done by M.S. The final manuscript is the result of the collaborative effort of both authors.

**Funding:** The University of Zululand Research and Innovation Office and the National Research Fund (NRF) sponsored this research.

**Acknowledgments:** The authors would like to gratefully acknowledge the University of Zululand Research and Innovation Office, the National Research Fund (NRF) for financial support during this study.

**Conflicts of Interest:** The authors declare no conflict of interest.

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
