# Peer review of "Does Urban and Peri-Urban Agriculture Contribute to Household Food Security? An Assessment of the Food Security Status of Households in Tongaat, eThekwini Municipality"

_sustainability, doi:10.3390/su11041082_

Round 1

Reviewer 1 Report

I have no more questions for authors.

Author Response

.

Reviewer 2 Report

The paper, that aims to assess the food security status of households that practised any form of UPA activities within the Tongaat peri-urban area, can be accepted after some minor revisions, consisting in careful rechecking of the statistical procedures performed.

Author Response

.

This manuscript is a resubmission of an earlier submission. The following is a list of the peer review reports and author responses from that submission.

Round 1

Reviewer 1 Report

I have a few comments to the reviewed manuscript:

The text must be considered and redrafted. It's very hard to read. Some parts are even not understandable.

Figure 1. I think that the drawing should be redrawn. It looks like it was incomplete.

Which spreadsheet was used to convert data. Authors should write in manuscript  exactly what program and what version was used.

The sum of values in some columns does not add up to 100.

Table 4 and 6 are illegible. Please reformat it.

Reviewer 2 Report

Dear Authors.. I think the topic of your research is very interesting.. but the I have a MAJOR issue about your approach, writing, statistical analysis, and concluding your results. I will explore with you my major concerns:

The text need major editing.. It is not easy to read.. and it is more suitable for a report, or at maximum a book chapter, but not a research article .. I will tell you why in my next comments. 

I have a MAJOR issue with your research question.. research objective: you stated that: "It is in this context that this paper seeks to assess the food security status of households practising any UPA activities against those households that did not practice any UPA activities in Tongaat peri-urban area of eThekwini Municipality".. If so.. I can give you the answer .. and I am sure anybody in the world can give you the answer for this question. This MUST be re-formulated. 

Introduction is very very general and not focused at all!!!

In many parts of the text you provided unnecessary explanations! such as: "Tabulation of HDDS was constructed by using a computer", what else would you use then papers!! and "measures of central tendency were computed (that is; the mean, range (minimum and maximum) and standard deviation)".. would you expect that researchers would not know what are the "measures of central tendency".. this is not a lecture note, don't forget!.. I am sorry to say that!!!

Figures!! First you used "Figure 1" twice for Figure 1 and Figure 2. Then in Figure 1, I would encourage you to show South Aftrica Map, and then the region, and it is fine to put a black dot showing your city. Then Figure 1 again, which should be Figure 2, you caption is "Theoretical framework"... Framework for what??? you need to explain?

Now let's go to the results section! Age "almost the same 21-72 vs. 22-70"!!! Family size "the same 10 vs 9"!! and schooling years "the same: 9.17 years vs. 9.16 years" , and the rest were so close to each other !!How, what is your Significant level, to conclude your stated results about the impact of age, family size and education, gender, etc.

Please, I recommend you to send your results to a statistician to test the statistical significnace of your results, if you can't do that.

I think you had a mistake in reading your data about the unemployment.. Please check that: in Table 1 you have an extra line of values. Please make sure if the unemployment rate is 56.1 vs 34.3, or 34.9 vs 65.7, or 50.4 vs 44.4? I will tell you what! I think you need to re-write this whole article and make the core objectives as what are the reasons that make urban people in this city practice  UPA activities or not practicing it??? and what are the consequences ?? as stating in Table 7 (for example). Again, to be fair, you have done great job in collecting the data, and your topic is very important, but you have chosen the wrong research objective and questions, and the way you presented and discussed the results needs a lot of work. 

Interestingly : your conclusion was in the right direction: stating why would people go for UPA or not, and then should focus what benefit they can get and also the government get if they encourage them to go for UPA instead providing support "easy money makes people lazy!" 

Reviewer 3 Report

I think the paper is a great job whose best strength is the relationship between UPA and food security at the urban level, a topic often underlined in the scientific literature but for which there is little empirical evidence.

However, I asked for the possibility of highlighting through which mechanisms the positive relationship between UPA and HDDS and HFIAS is realized.

The statistical relationship is clear, but why do you come up with this relationship?

For example, why in the families analyzed there is a greater share of self-consumption? Another reason could be the type of crops dedicated to fresh consumption rather than industrial? Or, and could the motivation be in company work? These clarifications would make the article more interesting.

I hope I have made my review more effective, but I remain at your disposal for further details.

Reviewer 4 Report

The general layout of the manuscript does not seem to reflect the generally used standard. Paragraph 2.1 should have its own autonomy, becoming Paragraph 2. Thus the methodology should become Par. 3.

In the Methology section, the authors dwell in explanations of basic concepts that however are of common understanding of the reader of the scientific journal and that therefore should be eliminated or otherwise mentioned in the notes.

The approach of the methodology seems very weak, focusing only on a descriptive and exploratory analysis. The methodology should therefore be reviewed.